# Advantage-Weighted Regression: Simple and Scalable Off-Policy Reinforcement Learning

## Abstract

In this work, we aim to develop a simple and scalable reinforcement learning algorithm that uses standard supervised learning methods as subroutines, while also being able to leverage off-policy data. Our proposed approach, which we refer to as *advantage-weighted regression* (AWR), consists of two standard supervised learning steps: one to regress onto target values for a value function, and another to regress onto weighted target actions for the policy. The method is simple and general, can accommodate continuous and discrete actions, and can be implemented in just a few lines of code on top of standard supervised learning methods. We provide a theoretical motivation for AWR and analyze its properties when incorporating off-policy data from experience replay. We evaluate AWR on a suite of standard OpenAI Gym benchmark tasks, and show that it achieves competitive performance compared to a number of well-established state-of-the-art RL algorithms. AWR is also able to acquire more effective policies than most off-policy algorithms when learning from purely static datasets with no additional environmental interactions. Furthermore, we demonstrate our algorithm on challenging continuous control tasks with highly complex simulated characters. (Video[1])

## 1 Introduction

Model-free reinforcement learning can be a general and effective methodology for training agents to acquire sophisticated behaviors with minimal assumptions on the underlying task. However, RL algorithms can be substantially more complex to implement and tune than standard supervised learning methods. Arguably the simplest reinforcement learning methods are policy gradient algorithms (Sutton et al., 2000), which directly differentiate the expected return and perform gradient ascent. Unfortunately, these methods can be notoriously unstable and are typically on-policy, often requiring a substantial number of samples to learn effective behaviors. Our goal is to develop an RL algorithm that is simple, easy to implement, and can readily incorporate off-policy data.

In this work, we propose advantage-weighted regression (AWR), a simple off-policy algorithm for model-free RL. Each iteration of the AWR algorithm simply consists of two supervised regression steps: one for training a value function baseline via regression onto cumulative rewards, and another for training the policy via weighted regression. The complete algorithm is shown in Algorithm 1. AWR can accommodate continuous and discrete actions, and can be implemented in just a few lines of code on top of standard supervised learning methods. Despite its simplicity, we find that AWR achieves competitive results when compared to commonly used on-policy and off-policy RL algorithms, and can effectively incorporate fully off-policy data, which has been a challenge for other RL algorithms. Our derivation presents an interpretation of AWR as a constrained policy optimization procedure, and provides a theoretical analysis of the use of off-policy data with experience replay.

We first revisit the original formulation of reward-weighted regression (RWR) (Peters & Schaal, 2007), an on-policy RL method that utilizes supervised learning to perform policy updates, and then propose a number of new design decisions that significantly improve performance on a suite of standard control benchmark tasks. We then provide a theoretical analysis of AWR, including the capability to incorporate off-policy data with experience replay. Although the design of AWR involves only a few simple design decisions, we show experimentally that these additions provide for a large improvement over previous methods for regression-based policy search, such as RWR,

---

[1]Supplementary video: sites.google.com/view/awr-supp/

while also being substantially simpler than more modern methods, such as MPO (Abdolmaleki et al., 2018b). We show that AWR achieves competitive performance when compared to several well-established state-of-the-art on-policy and off-policy algorithms.

## 2 PRELIMINARIES

In reinforcement learning, the objective is to learn a policy that maximizes an agent's expected return. At each time step $t$, the agent observes the state of the environment $\mathbf{s}_t$, and samples an action from a policy $\mathbf{a}_t \sim \pi(\mathbf{a}_t|\mathbf{s}_t)$. The agent then applies that action, which results in a new state $\mathbf{s}_{t+1}$ and a scalar reward $r_t = r(\mathbf{s}_t, \mathbf{a}_t)$. The goal is to learn a policy that maximizes the expected return $J(\pi)$,

$$J(\pi) = \mathbb{E}_{\tau \sim p_\pi(\tau)} \left[ \sum_{t=0}^{\infty} \gamma^t r_t \right] = \mathbb{E}_{\mathbf{s} \sim d_\pi(\mathbf{s}), a \sim \pi(\mathbf{a}|\mathbf{s})} \left[ r(\mathbf{s}, \mathbf{a}) \right], \tag{1}$$

where $p_\pi(\tau)$ represents the likelihood of a trajectory $\tau = \{(\mathbf{s}_0, \mathbf{a}_0, r_0), (\mathbf{s}_1, \mathbf{a}_1, r_1), ...\}$ under a policy $\pi$, and $\gamma \in [0, 1)$ is the discount factor. $d_\pi(\mathbf{s}) = \sum_{t=0}^{\infty} \gamma^t p(\mathbf{s}_t = \mathbf{s}|\pi)$ represents the unnormalized discounted state distribution induced by the policy $\pi$ (Sutton & Barto, 1998), and $p(\mathbf{s}_t = \mathbf{s}|\pi)$ is the likelihood of the agent being in state $\mathbf{s}$ after following $\pi$ for $t$ timesteps.

Our proposed AWR algorithm builds on ideas from reward-weighted regression (RWR) (Peters & Schaal, 2007), a policy search algorithm based on an expectation-maximization framework. At each iteration, the E-step constructs an estimate of the optimal policy according to $\pi^*(\mathbf{a}|\mathbf{s}) \propto \pi_k(\mathbf{a}|\mathbf{s}) \exp(\mathcal{R}_{\mathbf{s},\mathbf{a}}/\beta)$, where $\pi_k$ represents the policy at the $k$th iteration, $\mathcal{R}_{\mathbf{s},\mathbf{a}} = \sum_{t=0}^{\infty} \gamma^t r_t$ is the return, and $\beta > 0$ is a temperature parameter. Then the M-step projects $\pi^*$ onto the space of parameterized policies by solving a supervised regression problem:

$$\pi_{k+1} = \arg \max_\pi \mathbb{E}_{\mathbf{s} \sim d_{\pi_k}(\mathbf{s})} \mathbb{E}_{\mathbf{a} \sim \pi_k(\mathbf{a}|\mathbf{s})} \left[ \log \pi(\mathbf{a}|\mathbf{s}) \exp \left( \frac{1}{\beta} \mathcal{R}_{\mathbf{s},\mathbf{a}} \right) \right]. \tag{2}$$

The RWR update can be interpreted as fitting a new policy $\pi_{k+1}$ to samples from the current policy $\pi_k$, where the likelihood of each action is weighted by the exponentiated return for that action.

## 3 ADVANTAGE-WEIGHTED REGRESSION

In this work, we present advantage-weighted regression (AWR), a simple off-policy RL algorithm based on reward-weighted regression. We first provide an overview of the AWR algorithm, and then describe its theoretical motivation and analyze its properties. The AWR algorithm is summarized in Algorithm 1. Each iteration $k$ of AWR consists of the following simple steps. First, the current policy $\pi_k(\mathbf{a}|\mathbf{s})$ is used to sample a batch of trajectories $\{\tau_i\}$ that are then stored in the replay buffer $\mathcal{D}$, which is structured as a first-in first-out (FIFO) queue (Mnih et al., 2015). Then, a value function $V_k^{\mathcal{D}}(\mathbf{s})$ is fitted to all trajectories in the replay buffer $\mathcal{D}$, which can be done with simple Monte Carlo return estimates $\mathcal{R}_{\mathbf{s},\mathbf{a}}^{\mathcal{D}} = \sum_{t=0}^{T} \gamma^t r_t$. Finally, the same buffer is used to fit a new policy using *advantage-weighted* regression, where each state-action pair in the buffer is weighted according to the exponentiated advantage $\exp(\frac{1}{\beta} A^{\mathcal{D}}(\mathbf{s}, \mathbf{a}))$, with the advantage given by $A^{\mathcal{D}}(\mathbf{s}, \mathbf{a}) = \mathcal{R}_{\mathbf{s},\mathbf{a}}^{\mathcal{D}} - V^{\mathcal{D}}(\mathbf{s})$ and $\beta$ is a hyperparameter. In the following subsections, we first motivate AWR as a constrained policy search problem, and then extend our analysis to incorporate experience replay.

### 3.1 DERIVATION

In this section, we derive the AWR algorithm as an approximate optimization of a constrained policy search problem. Our goal is to find a policy that maximizes the expected *improvement* $\eta(\pi) = J(\pi) - J(\mu)$ over a sampling policy $\mu(\mathbf{a}|\mathbf{s})$. We first derive AWR for the setting where the sampling policy is a single Markovian policy. Then, in the next section, we extend our result to data from multiple policies, as in the case of experience replay. The expected improvement $\eta(\pi)$ can be expressed in terms of the advantage $A^\mu(\mathbf{s}, \mathbf{a}) = \mathcal{R}_{\mathbf{s},\mathbf{a}}^\mu - V^\mu(\mathbf{s})$ with respect to $\mu$ (Kakade & Langford, 2002; Schulman et al., 2015):

$$\eta(\pi) = \mathbb{E}_{\mathbf{s} \sim d_\pi(\mathbf{s})} \mathbb{E}_{\mathbf{a} \sim \pi(\mathbf{a}|\mathbf{s})} \left[ \mathcal{R}_{\mathbf{s},\mathbf{a}}^\mu - V^\mu(\mathbf{s}) \right], \tag{3}$$

where $\mathcal{R}_{\mathbf{s},\mathbf{a}}^\mu$ denotes the return obtained by performing action $\mathbf{a}$ in state $\mathbf{s}$ and following $\mu$ for the following timesteps, and $V^\mu(\mathbf{s}) = \int_a \mu(\mathbf{a}|\mathbf{s}) \mathcal{R}_\mathbf{s}^\mathbf{a} d\mathbf{a}$ corresponds to the value function of $\mu$. This objective differs from the ones used in the derivations of related algorithms, such as RWR and

---

**Algorithm 1** Advantage-Weighted Regression

---

1: $\pi_1 \leftarrow$ random policy
2: $\mathcal{D} \leftarrow \emptyset$
3: **for** iteration $k = 1, ..., k_{\max}$ **do**
4:      add trajectories $\{\tau_i\}$ sampled via $\pi_k$ to $\mathcal{D}$
5:      $V_k^{\mathcal{D}} \leftarrow \arg\min_V \mathbb{E}_{\mathbf{s,a}\sim\mathcal{D}} \left[ \left|\left| \mathcal{R}_{\mathbf{s,a}}^{\mathcal{D}} - V(\mathbf{s}) \right|\right|^2 \right]$
6:      $\pi_{k+1} \leftarrow \arg\max_\pi \mathbb{E}_{\mathbf{s,a}\sim\mathcal{D}} \left[ \log \pi(\mathbf{a}|\mathbf{s}) \exp\left( \frac{1}{\beta} \left( \mathcal{R}_{\mathbf{s,a}}^{\mathcal{D}} - V_k^{\mathcal{D}}(\mathbf{s}) \right) \right) \right]$
7: **end for**

---

REPS (Peters & Schaal, 2007; Peters et al., 2010; Abdolmaleki et al., 2018b), which maximize the expected return $J(\pi)$ instead of the expected improvement. The expected improvement directly gives rise to an objective that involves the advantage. We will see later that this yields a policy update that differ in a subtle but important way from standard RWR. As we show in our experiments, this difference results in a large empirical improvement.

The objective in Equation 3 can be difficult to optimize due to the dependency between $d_\pi(\mathbf{s})$ and $\pi$, as well as the need to collect samples from $\pi$. Following Schulman et al. (2015), we can instead optimize an approximation $\hat{\eta}(\pi)$ of $\eta(\pi)$ using the state distribution of $\mu$:

$$\hat{\eta}(\pi) = \mathbb{E}_{\mathbf{s}\sim d_\mu(\mathbf{s})} \mathbb{E}_{\mathbf{a}\sim\pi(\mathbf{a}|\mathbf{s})} \left[ \mathcal{R}_{\mathbf{s,a}}^\mu - V^\mu(\mathbf{s}) \right]. \tag{4}$$

Here, $\hat{\eta}(\pi)$ matches $\eta(\pi)$ to first order (Kakade & Langford, 2002), and provides a good estimate of $\eta$ if $\pi$ and $\mu$ are close in terms of the KL-divergence (Schulman et al., 2015). Using this objective, we can formulate the following *constrained* policy search problem:

$$\arg\max_\pi \quad \int_\mathbf{s} d_\mu(\mathbf{s}) \int_\mathbf{a} \pi(\mathbf{a}|\mathbf{s}) \left[ \mathcal{R}_{\mathbf{s,a}}^\mu - V^\mu(\mathbf{s}) \right] \, d\mathbf{a} \, d\mathbf{s} \tag{5}$$

$$\text{s.t.} \quad \int_\mathbf{s} d_\mu(\mathbf{s}) \mathrm{D_{KL}}\left( \pi(\cdot|\mathbf{s}) || \mu(\cdot|\mathbf{s}) \right) d\mathbf{s} \leq \epsilon. \tag{6}$$

The constraint in Equation 6 ensures that the new policy $\pi$ is close to the data distribution of $\mu$, and therefore the surrogate objective $\hat{\eta}(\pi)$ remains a reasonable approximation to $\eta(\pi)$. We refer the reader to Schulman et al. (2015) for a detailed derivation and an error bound.

We can derive AWR as an approximate solution to this constrained optimization. This derivation follows a similar procedure as Peters et al. (2010), and begins by forming the Lagrangian of the optimization problem presented above,

$$\mathcal{L}(\pi,\beta) = \int_\mathbf{s} d_\mu(\mathbf{s}) \int_\mathbf{a} \pi(\mathbf{a}|\mathbf{s}) \left[ \mathcal{R}_{\mathbf{s,a}}^\mu - V^\mu(\mathbf{s}) \right] d\mathbf{a} \, d\mathbf{s} + \beta \left( \epsilon - \int_\mathbf{s} d_\mu(\mathbf{s}) \mathrm{D_{KL}}\left( \pi(\cdot|\mathbf{s}) || \mu(\cdot|\mathbf{s}) \right) d\mathbf{s} \right), \tag{7}$$

where $\beta$ is a Lagrange multiplier. Differentiating $\mathcal{L}(\pi, \beta)$ with respect to $\pi(\mathbf{a}|\mathbf{s})$ and solving for the optimal policy $\pi^*$ results in the following expression for the optimal policy

$$\pi^*(\mathbf{a}|\mathbf{s}) = \frac{1}{Z(\mathbf{s})} \mu(\mathbf{a}|\mathbf{s}) \exp\left( \frac{1}{\beta} \left( \mathcal{R}_{\mathbf{s,a}}^\mu - V^\mu(\mathbf{s}) \right) \right), \tag{8}$$

with $Z(\mathbf{s})$ being the partition function. A detailed derivation is available in Appendix A. If $\pi$ is represented by a function approximator (e.g., a neural network), a new policy can be obtained by projecting $\pi^*$ onto the manifold of parameterized policies,

$$\arg\min_\pi \quad \mathbb{E}_{\mathbf{s}\sim\mathcal{D}} \left[ \mathrm{D_{KL}}\left( \pi^*(\cdot|\mathbf{s}) || \pi(\cdot|\mathbf{s}) \right) \right] \tag{9}$$

$$= \arg\max_\pi \quad \mathbb{E}_{\mathbf{s}\sim d_\mu(\mathbf{s})} \mathbb{E}_{\mathbf{a}\sim\mu(\mathbf{a}|\mathbf{s})} \left[ \log \pi(\mathbf{a}|\mathbf{s}) \exp\left( \frac{1}{\beta} \left( \mathcal{R}_{\mathbf{s,a}}^\mu - V^\mu(\mathbf{s}) \right) \right) \right]. \tag{10}$$

While this derivation for AWR largely follows the derivations used in prior work (Peters et al., 2010; Abdolmaleki et al., 2018b), our expected improvement objective introduces a baseline $V^\mu(\mathbf{s})$ to the policy update, which as we show in our experiments, is a crucial component for an effective algorithm. A similar advantage-weighting scheme has been previously used for fitted Q-iteration (Neumann & Peters, 2009), where the policy is given by $\pi(\mathbf{a}|\mathbf{s}) = \frac{1}{Z(\mathbf{s})} \exp\left( \left( \mathcal{R}_{\mathbf{s,a}}^\mu - V^\mu(\mathbf{s}) \right) / \beta \right)$. In this definition, the likelihood of an action does not depend on the sampling distribution, and therefore does not enforce a trust region with respect to $\mu$.

## 3.2 Experience Replay and Off-Policy Learning

A crucial design decision of AWR is the choice of sampling policy $\mu(\mathbf{a}|\mathbf{s})$. Standard implementations of RWR are typically on-policy, where the sampling policy is selected to be the current policy $\mu(\mathbf{a}|\mathbf{s}) = \pi_k(\mathbf{a}|\mathbf{s})$ at iteration $k$. This can be sample inefficient, as data collected at each iteration are discarded after a single update iteration. Importance sampling can be incorporated into RWR to reuse data from previous iterations, but at the cost of larger variance (Kober & Peters, 2009). Instead, we can improve sample efficiency of AWR by incorporating experience replay and explicitly accounting for training data from a mixture of multiple past policies. As described in Algorithm 1, at each iteration, AWR collects a batch of data using the latest policy $\pi_k$, and then stores this data in a replay buffer $\mathcal{D}$, which also contains data collected from previous policies $\{\pi_1, \cdots, \pi_k\}$. The value function and policy are then updated using samples drawn from $\mathcal{D}$. This replay strategy is analogous to modeling the sampling policy as a mixture of policies from previous iterations $\mu_k(\tau) = \sum_{i=1}^{k} w_i \pi_i(\tau)$, where $\pi_i(\tau) = p(\tau|\pi_i)$ represents the likelihood of a trajectory $\tau$ under a policy $\pi_i$ from the $i$th iteration, and the weight $w_i$ specify the probability of selecting $\pi_i$.

We now extend the derivation from the previous section to the off-policy setting with experience replay, and show that Algorithm 1 indeed optimizes the expected improvement over a sampling policy modeled by the replay buffer. Given a replay buffer consisting of trajectories from past policies, the joint state-action distribution of $\mu$ is given by $\mu(\mathbf{s}, \mathbf{a}) = \sum_{i=1}^{k} w_i d_{\pi_i}(\mathbf{s}) \pi_i(\mathbf{a}|\mathbf{s})$, and similarly for the marginal state distribution $d_\mu(\mathbf{s}) = \sum_{i=1}^{k} w_i d_{\pi_i}(\mathbf{s})$. The *expected improvement* can now be expressed with respect to the set of sampling policies in the replay buffer,

$$\eta(\pi) = J(\pi) - \sum_i w_i J(\pi_i) = \mathbb{E}_{\mathbf{s} \sim d_\pi(\mathbf{s})} \mathbb{E}_{\mathbf{a} \sim \pi(\mathbf{a}|\mathbf{s})} \left[ \sum_i w_i A^{\pi_i}(\mathbf{s}, \mathbf{a}) \right], \tag{11}$$

where $A^{\pi_i}(\mathbf{s}, \mathbf{a}) = \mathcal{R}_{\mathbf{s}, \mathbf{a}}^{\pi_i} - V^{\pi_i}(\mathbf{s})$ is the advantage with respect to each sampling policy. In Appendix B, we show that the update procedure in Algorithm 1 optimizes the following objective:

$$\arg\max_\pi \sum_{i=1}^{k} w_i \left( \mathbb{E}_{\mathbf{s} \sim d_{\pi_i}(\mathbf{s})} \mathbb{E}_{\mathbf{a} \sim \pi(\mathbf{a}|\mathbf{s})} \left[ A^{\pi_i}(\mathbf{s}, \mathbf{a}) \right] \right) \tag{12}$$

$$\text{s.t.} \quad \mathbb{E}_{\mathbf{s} \sim d_\mu(\mathbf{s})} \left[ D_{\mathrm{KL}} \left( \pi(\cdot|\mathbf{s}) || \mu(\cdot|\mathbf{s}) \right) \right] \leq \epsilon, \tag{13}$$

where $\mu(\mathbf{a}|\mathbf{s}) = \frac{\mu(\mathbf{s}, \mathbf{a})}{d_\mu(\mathbf{s})} = \frac{\sum_i w_i d_{\pi_i}(\mathbf{s}) \pi_i(\mathbf{a}|\mathbf{s})}{\sum_j w_j d_{\pi_j}(\mathbf{s})}$ represents the conditional action distribution defined by the replay buffer. This objective can be solved via the Lagrangian to yield the following update:

$$\arg\max_\pi \sum_{i=1}^{k} w_i \, \mathbb{E}_{\mathbf{s} \sim d_{\pi_i}(\mathbf{s})} \mathbb{E}_{\mathbf{a} \sim \pi_i(\mathbf{a}|\mathbf{s})} \left[ \log \pi(\mathbf{a}|\mathbf{s}) \exp \left( \frac{1}{\beta} \left( \mathcal{R}_{\mathbf{s}, \mathbf{a}}^{\pi_i} - \frac{\sum_j w_j d_{\pi_j}(\mathbf{s}) V^{\pi_j}(\mathbf{s})}{\sum_j w_j d_{\pi_j}(\mathbf{s})} \right) \right) \right], \tag{14}$$

where the expectations can be approximated by simply sampling from $\mathcal{D}$ following Line 6 of Algorithm 1. A detailed derivation is available in Appendix B. Note, the baseline in the exponent now consists of an average of the value functions of the different policies. This *mean* value function $\bar{V}(\mathbf{s})$ can be fitted by simply sampling from the replay buffer following Line 5 of Algorithm 1,

$$\bar{V} = \arg\min_V \sum_i w_i \mathbb{E}_{\mathbf{s}, \sim d_{\pi_i}(\mathbf{s}), \mathbf{a} \sim \pi_i(\mathbf{a}|\mathbf{s})} \left[ ||\mathcal{R}_{\mathbf{s}, \mathbf{a}}^{\pi_i} - V(\mathbf{s})||^2 \right]. \tag{15}$$

The optimal solution $\bar{V}(\mathbf{s}) = \frac{\sum_i w_i d_{\pi_i}(\mathbf{s}) V^{\pi_i}(\mathbf{s})}{\sum_j w_j d_{\pi_j}(\mathbf{s})}$ is exactly the baseline in Equation 14.

## 3.3 Implementation Details

Finally, we discuss several important design decisions for a practical implementation of AWR. Monte Carlo estimates can be used to approximate the expected return $\mathcal{R}_{\mathbf{s}, \mathbf{a}}^{\mathcal{D}}$, but this can result in a high-variance estimate. Instead, we approximate $\mathcal{R}_{\mathbf{s}, \mathbf{a}}^{\mathcal{D}}$ using TD($\lambda$) to obtain a lower-variance estimate (Sutton & Barto, 1998). TD($\lambda$) is applied by bootstrapping with the value function $V_{k-1}^{\mathcal{D}}(\mathbf{s})$ from the previous iteration. To set the value of the Lagrange multiplier $\beta$, we found that a simple adaptive heuristic of setting $\beta$ to the standard deviation of all advantage values $\sigma_A$ in the replay buffer works well in practice. This is akin to the advantage normalization technique commonly used in implementations of algorithms such as PPO (Dhariwal et al., 2017). Details are available in Appendix C.

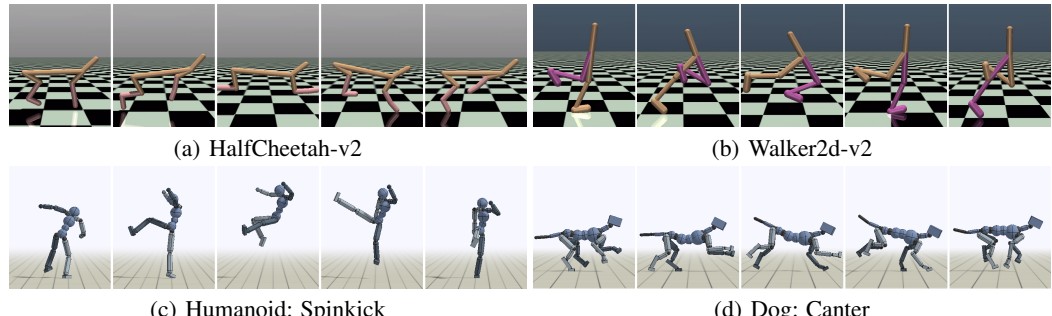

(a) HalfCheetah-v2

(b) Walker2d-v2

(c) Humanoid: Spinkick

(d) Dog: Canter

Figure 1: Snapshots of AWR policies trained on OpenAI Gym and motion imitation tasks. Our simple algorithm learns effective policies for a diverse suite of control tasks.

The weights $\omega_{\mathbf{s,a}}^{\mathcal{D}} = \exp\left(\frac{1}{\beta}\left(\mathcal{R}_{\mathbf{s,a}}^{\mathcal{D}} - V^{\mathcal{D}}(\mathbf{s})\right)\right)$ used to update the policy can occasionally assume excessively large values, which causes gradients to explode. Therefore, we apply weight clipping $\hat{\omega}_{\mathbf{s,a}}^{\mathcal{D}} = \min\left(\omega_{\mathbf{s,a}}^{\mathcal{D}}, \omega_{\max}\right)$ with a threshold $\omega_{\max}$ to prevent exploding weights.

## 4 RELATED WORK

Existing RL methods can be broadly categorized into on-policy and off-policy algorithms (Sutton & Barto, 1998). On-policy algorithms generally update the policy using data collected from the same policy. A popular class of on-policy algorithms is policy gradient methods (Williams, 1992; Sutton et al., 2000), which can be effective for a diverse array of complex tasks (Heess et al., 2017; Pathak et al., 2017; Peng et al., 2018; Rajeswaran et al., 2018). However, on-policy algorithms are typically data inefficient. Off-policy algorithms improve sample efficiency by enabling training using data from other sources, such as data from different agents or data from previous iterations of the algorithm. Importance sampling is a simple strategy for off-policy learning (Sutton & Barto, 1998; Meuleau et al., 2000; Hachiya et al., 2009), but can introduce optimization instabilities due to the large variance of the importance sampling estimator. Dynamic programming methods based on Q-function learning can also leverage off-policy data (Precup et al., 2001; Mnih et al., 2015; Lillicrap et al., 2016; Gu et al., 2016; Haarnoja et al., 2018b). But these methods can be notoriously unstable, and in practice, require a variety of stabilization techniques (Hasselt et al., 2016; Wang et al., 2016; Munos et al., 2016; Hessel et al., 2017; Fujimoto et al., 2018; Fu et al., 2019). Furthermore, it can be difficult to apply these methods to fully off-policy data, where an agent is unable to collect additional environmental interactions (Fujimoto et al., 2019; Kumar et al., 2019).

Policy search can also be formulated under an expectation-maximization framework (Peters et al., 2010; Neumann, 2011; Abdolmaleki et al., 2018b), an early example of which is reward-weighted regression (RWR) (Peters & Schaal, 2007). RWR presents a simple on-policy RL algorithm that casts policy search as a supervised regression problem. A similar algorithm, relative entropy policy search (REPS) (Peters et al., 2010), can also be derived from the dual formulation of a constrained policy search problem. RWR has a number appealing properties: it has a very simple update rule, and since each iteration corresponds to supervised learning, it can be more stable and easier to implement than many of the previously mentioned RL methods. Despite these advantages, RWR has not been shown to be an effective when combined with neural networks (Schulman et al., 2015; Duan et al., 2016). In this work, we propose a number of modifications to the formulation of RWR to produce an effective off-policy deep RL algorithm, while still retaining much of the simplicity of RWR.

The optimization problem being solved in AWR is similar to REPS (Peters et al., 2010), but REPS optimizes the expected return instead of the expected improvement. The weights in REPS also contains a Bellman error term that resembles advantages, but are computed using a linear value function derived from a feature matching constraint. Learning the REPS value function involves minimization of a dual function, which is a complex function of the Bellman error, while the value function in AWR can be learned with simple supervised regression. More recently, Abdolmaleki et al. (2018b) proposed MPO, a deep RL variant of REPS, which applies a partial EM algorithm for policy optimization. The method first fits a Q-function of the current policy via bootstrapping, and then performs a policy improvement step with respect to this Q-function. MPO uses off-policy data for training a Q-function and employs Retrace($\lambda$) for off-policy correction (Munos et al., 2016). In

Figure 2: Learning curves of the various algorithms when applied to OpenAI Gym tasks. Results are averaged across 10 random seeds. AWR is generally competitive with the best current methods.

| Task | TRPO | PPO | DDPG | TD3 | SAC | LAWER | RWR | AWR (Ours) |
|---|---|---|---|---|---|---|---|---|
| Ant-v2 | $2901 \pm 85$ | $4884 \pm 1249$ | $72 \pm 1550$ | $5997 \pm 765$ | $\mathbf{7500 \pm 353}$ | $2240 \pm 497$ | $1183 \pm 176$ | $5372 \pm 163$ |
| HalfCheetah-v2 | $3302 \pm 428$ | $7617 \pm 185$ | $10563 \pm 382$ | $12324 \pm 1549$ | $\mathbf{16223 \pm 964}$ | $4596 \pm 2331$ | $2075 \pm 370$ | $9192 \pm 157$ |
| Hopper-v2 | $1880 \pm 337$ | $2514 \pm 726$ | $855 \pm 282$ | $2794 \pm 15$ | $2757 \pm 658$ | $1830 \pm 553$ | $605 \pm 114$ | $\mathbf{3498 \pm 167}$ |
| Humanoid-v2 | $552 \pm 9$ | $4668 \pm 1153$ | $4382 \pm 423$ | $4738 \pm 93$ | $\mathbf{6296 \pm 332}$ | $108 \pm 386$ | $509 \pm 18$ | $6159 \pm 274$ |
| LunarLander-v2 | $104 \pm 94$ | $121 \pm 49$ | – | – | – | – | $185 \pm 23$ | $\mathbf{229 \pm 2}$ |
| Walker2d-v2 | $2765 \pm 168$ | $5036 \pm 934$ | $401 \pm 470$ | $4779 \pm 803$ | $\mathbf{6210 \pm 511}$ | $2502 \pm 388$ | $406 \pm 64$ | $5813 \pm 483$ |

Table 1: Final returns for different algorithms on the OpenAI Gym tasks, with $\pm$ corresponding to one standard deviation of the average return across 10 random seeds. In terms of final performance, AWR is generally competitive with prior methods.

contrast, AWR is simpler, as it can simply fit a value function to the observed returns in a replay buffer, and performs weighted supervised regression on the actions to fit the policy. Oh et al. (2018) proposed self-imitation learning (SIL), which augments policy gradient algorithms with an auxiliary behaviour cloning loss to reuse samples from past experiences. Unlike SIL, AWR is a standalone algorithm, and does not need to be combined with an auxiliary RL algorithm. Neumann & Peters (2009) proposed LAWER, a kernel-based fitted Q-iteration algorithm where the Bellman error is weighted by the normalized advantage of each state-action pair. This was then followed by a soft-policy improvement step. Similar to Neumann & Peters (2009), AWR also uses exponentiated advantages, but LAWER's definition of the policy is different from the one in AWR and does not enforce a trust region constraint. Furthermore, AWR does not perform fitted Q-iteration, and instead utilizes off-policy data in a simple constrained policy search procedure. Wang et al. (2018) applied a similar advantage-weighting scheme for imitation learning, but the method was not demonstrated for the RL setting. In this work, we propose several design decisions that are vital for an effective RL algorithm. We also provide a theoretical analysis of AWR when combined with experience replay, and show that the algorithm optimizes the expected improvement with respect to a mixture of policies modeled by a replay buffer.

## 5 EXPERIMENTS

Our experiments aim to comparatively evaluate the performance of AWR with commonly used on-policy and off-policy deep RL algorithms. We evaluate our method on the OpenAI Gym benchmarks (Brockman et al., 2016), consisting of discrete and continuous control tasks. We also evaluate our method on complex motion imitation tasks with high-dimensional simulated characters. We then demonstrate the effectiveness of AWR on fully off-policy learning, by training on static datasets of demonstrations from demo policies. Behaviors learned by the policies are best seen in the supplementary video[1]. Code for our implementation of AWR is available at sites.google.com/view/awr-supp/. Detailed hyperparameter settings are provided in Appendix C.

### 5.1 BENCHMARKS

We compare AWR to a number of state-of-the-art RL algorithms, including on-policy algorithms, such as TRPO (Schulman et al., 2015) and PPO (Schulman et al., 2017), off-policy algorithms, such as DDPG (Lillicrap et al., 2016), TD3 (Fujimoto et al., 2018), and SAC (Haarnoja et al., 2018a), as well as RWR (Peters & Schaal, 2007) and LAWER (Neumann & Peters, 2009).[2] TRPO, PPO, and DDPG use the implementations from OpenAI baselines (Dhariwal et al., 2017). TD3 and SAC use the implementations from Fujimoto et al. (2018) and Haarnoja et al. (2018a). RWR and LAWER are implemented following the descriptions in Peters & Schaal (2007) and Neumann & Peters (2009), but neural networks are used instead of kernel-based approximators.

---

[2]While we attempted to compare to MPO (Abdolmaleki et al., 2018b), we were unable to find or implement a version of the algorithm that reproduces the results comparable to those reported by Abdolmaleki et al. (2018b).

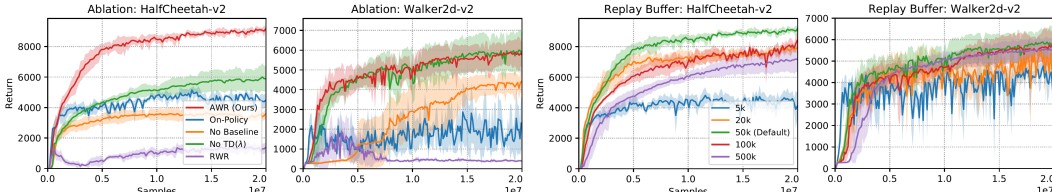

Figure 3: **Left:** Learning curves comparing AWR with various components removed. Each component contributes to performance improvements. **Right:** Learning curves comparing AWR with different capacity replay buffers. AWR remains stable with large buffers containing primarily off-policy data from past iterations.

Snapshots of the AWR policies are shown in Figure 1. Figure 2 shows learning curves comparing the different algorithms, and Table 1 summarizes the average returns of the final policies across 10 training runs initialized with different random seeds. Due to the slow wall-clock times of TD3 and SAC, some training runs did not have sufficient time to collect as many samples as other algorithms. Overall, AWR shows competitive performance with the state-of-the-art deep RL algorithms. It is competitive with on-policy methods, such as TRPO and PPO, in both sample efficiency and asymptotic performance. While it is not yet as sample efficient as current state-of-the-art off-policy methods, such SAC and TD3, it is able to achieve a comparable asymptotic performance on most tasks. RWR tends to perform poorly on these tasks, which suggests that, the particular modifications from AWR are critical. AWR also significantly outperforms LAWER across the various tasks. Though both methods use a similar advantaged-weighting scheme, our design decisions for AWR produce a simpler and more effective algorithm.

## 5.2 ABLATION EXPERIMENTS

To determine the effects of various design decisions, we evaluate the performance of AWR when key components have been removed. The experiments include: an on-policy version of AWR (On-Policy), where updates use only data from the latest policy, a version of AWR without the baseline $V(\mathbf{s})$ (No Baseline), and a version that uses Monte Carlo return estimates instead of TD($\lambda$) (No TD($\lambda$)). The effects of these components are illustrated in Figure 3. Overall, these design decisions appear to be vital for an effective algorithm, with the most crucial components being the use of experience replay and a baseline. Updates using only on-policy data can lead to instabilities and noticeable degradation in performance, which may be due to overfitting on a smaller dataset. Removing the baseline also noticeably hampers performance. Using simple Monte Carlo return estimates instead of TD($\lambda$) seems to be a viable alternative, and the algorithm still achieves competitive performance on some tasks. When combined, these different components yield substantial performance gains over standard RWR.

To further evaluate the effect of experience replay, we compare policies trained using replay buffer with different capacities. Figure 3 illustrates the learning curves for buffers of size 5k, 20k, 50k, 100k, and 500k, with 50k being the default buffer size in our experiments. The size of the replay buffer appears to have a significant impact on overall performance. Smaller buffer sizes can result in instabilities during training, which again may be an effect of overfitting to a smaller dataset. As the buffer size increases, AWR remains stable even when the dataset is dominated by off-policy data from previous iterations. In fact, AWR appears more stable with larger replay buffers, but progress can also become slower. Since the sampling policy $\mu(\mathbf{a}|\mathbf{s})$ is modeled by the replay buffer, a larger buffer can limit the rate at which $\mu$ changes by maintaining older data for more iterations.

## 5.3 MOTION IMITATION

In this section, we show that AWR can also solve high-dimensional tasks with complex simulated characters, including a 34 DoF humanoid and 64 DoF dog. The objective of the tasks is to imitate reference motion clips recorded using mocap. The experimental setup follows the framework proposed by Peng et al. (2018). The motions include walking and running (e.g. canter), as well as acrobatic skills, such as cartwheels and spinkicks. Figure 1 shows snapshots of the behaviors learned by the AWR. Table 2 and Figure 4 compare the performance of AWR to RWR and PPO. AWR performs well across the set of challenging skills, consistently achieving comparable or better performance than PPO. RWR struggles with controlling the humanoid, but exhibits stronger performance on the dog. This difference may be due to the more dynamic and acrobatic skills of the humanoid.

| Task | PPO | RWR | AWR (Ours) |
|---|---|---|---|
| Humanoid: Cartwheel | $0.76 \pm 0.02$ | $0.03 \pm 0.01$ | $\mathbf{0.78 \pm 0.07}$ |
| Humanoid: Spinkick | $0.70 \pm 0.02$ | $0.05 \pm 0.03$ | $\mathbf{0.77 \pm 0.04}$ |
| Dog: Canter | $0.76 \pm 0.03$ | $0.78 \pm 0.04$ | $\mathbf{0.86 \pm 0.01}$ |
| Dog: Trot | $\mathbf{0.86 \pm 0.01}$ | $\mathbf{0.86 \pm 0.01}$ | $\mathbf{0.86 \pm 0.03}$ |
| Dog: Turn | $0.75 \pm 0.02$ | $0.75 \pm 0.03$ | $\mathbf{0.82 \pm 0.03}$ |

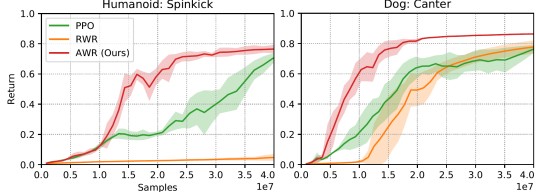

Table 2: Performance of algorithms on the motion imitation tasks. Returns are normalized between the minimum and maximum possible returns.

Figure 4: Learning curves on motion imitation tasks. On these challenging tasks, AWR generally learns faster than PPO and RWR.

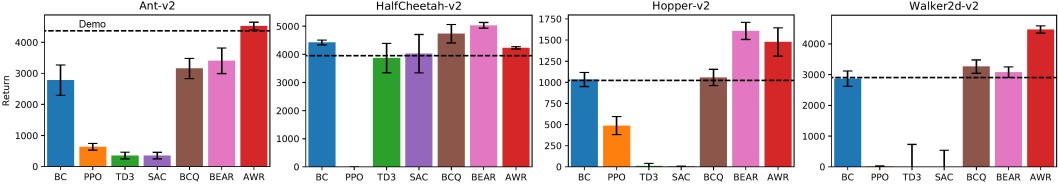

Figure 5: Performance of various algorithms on off-policy learning tasks with static datasets. AWR is able to learn policies that are comparable or better than the original demo policies.

## 5.4 OFF-POLICY LEARNING WITH STATIC DATASETS

Next, we evaluate AWR in a fully off-policy setting, where the algorithm is provided with a static dataset of experiences, and then tasked with learning the best possible policy without collecting any additional data. To evaluate our method, we use the off-policy tasks proposed by Kumar et al. (2019). The dataset consists of trajectories $\tau = \{(\mathbf{s}_0, \mathbf{a}_0, r_0), (\mathbf{s}_1, \mathbf{a}_1, r_1), ...\}$ from rollouts of a demo policy. Unlike standard imitation learning tasks, which only observes the states and actions from the demo policy, the dataset also records the reward at each step. The demo policies are trained using SAC on various OpenAI Gym tasks. A dataset of 1 million timesteps is collected for each task.

For AWR, we simply treat the dataset as the replay buffer $\mathcal{D}$ and directly apply the algorithm without any modifications. Figure 5 compares AWR to the original demo policy (Demo) and a behavioral cloning policy (BC). We also include comparisons to recent off-policy methods: batch-constrained Q-learning (BCQ) (Fujimoto et al., 2019) and bootstrapping error accumulation reduction (BEAR) (Kumar et al., 2019), which have shown strong performance on off-policy learning with static datasets. Note that both of these prior methods are modifications to existing off-policy RL methods, such as TD3 and SAC, which are already quite complex. In contrast, AWR is simple and requires no modifications for the fully off-policy setting. Despite not collecting any additional data, AWR is able to learn effective policies from these fully off-policy datasets, achieving comparable or better performance than the original demo policies. On-policy methods, such as PPO performs poorly in this off-policy setting. Q-function based methods, such as TD3 and SAC, can in principle handle off-policy data but tend to struggle in practice (Fujimoto et al., 2019; Kumar et al., 2019). Unlike Q-function based methods, AWR is less susceptible to issues from out-of-distribution actions as the policy is always trained on observed actions from the behaviour data (Kumar et al., 2019). AWR also shows comparable performance to BEAR and BCQ, which are specifically designed for this off-policy setting and introduce considerable algorithmic overhead.

## 6 DISCUSSION AND FUTURE WORK

We presented advantage-weighted regression, a simple off-policy reinforcement learning algorithm, where policy updates are performed using standard supervised learning methods. Despite its simplicity, our algorithm is able to solve challenging control tasks with complex simulated agents, and achieve competitive performance on standard benchmarks compared to a number of well-established RL algorithms. Our derivation introduces several new design decisions, and our experiments verify the importance of these components. AWR is also able to learn from fully off-policy datasets, demonstrating comparable performance to state-of-the-art off-policy methods. While AWR is effective for a diverse suite of tasks, it is not yet as sample efficient as the most efficient off-policy algorithms. We believe that exploring techniques for improving sample efficiency and performance on fully off-policy learning can open opportunities to deploy these methods in real world domains. A better theoretical understanding of the convergence properties of these algorithms, especially when combined with experience replay, could also be valuable for the development of future algorithms.

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
