# OpenReview forum: "Advantage-Weighted Regression: Simple and Scalable Off-Policy Reinforcement Learning"
_ICLR.cc/2021/Conference — Reject_

### Official Review · AnonReviewer1 · 2020-10-26
**Difference between MPO and AWR is not clear**

**Rating:** 6
**Confidence:** 3

**Review:**

This study presents a deep reinforcement learning method, Advantage-Weighted Regression (AWR). The policy update of AWR is constrained as in a similar manner as REPS (Peters et al., 2010). Although the benefit of AWR is not clear in the reinforcement learning tasks, AWR exhibits its advantages in the context of imitation learning and off-policy learning with static datasets.

The paper is well-organized and easy to follow. However, there are some unclear points. I would like to ask the authors to clarify the following points.

My main concern is the novelty of the method because the difference between MPO and AWR is not clear to me. In my understanding, both MPO and AWR solve the same optimization problem to satisfy the KL divergence constraint, and both MPO and AWR update the policy by maximizing the weighted log-likelihood. The difference I’m aware of is
1)	MPO uses the Q-function, while AWR uses the advantage function
2)	MPO uses Retrace, while AWR uses TD(\lambda)

The use of Retrace may not be important because Retrace can also be used for AWR. If we remove the baseline from AWR and uses Retrace instead of TD(\lambda), is it equivalent to MPO? Please correct me if I misunderstand anything.

Other questions:
- I do not clearly understand why AWR is suitable for off-policy learning with static datasets. Please provide the rationale.

- Why SAC and TD3 are not compared in the imitation learning tasks? I do not understand why only PPO and RWR were chosen as baselines. In addition, please describe the reason why the AWR is suitable for the imitation learning setting.

Minor points
- Although the “simplicity” of AWR is claimed several times, I’m not sure whether AWR is really simple because the difference between AWR and MPO is not big.

---

> ### Author Response · Authors · 2020-11-16
> **Response to R1**
>
> Re: Difference wrt to MPO
> A crucial difference between MPO and AWR that allows AWR to work well with offline datasets is the actions used to update the policy. During policy updates for MPO, new actions are sampled from the policy, which may be different from the actions observed during the rollouts. These actions are then scored by the Q function and used to update the policy. However, one problem with this update scheme is that it can cause out-of-distribution errors when querying the Q function with actions that were not observed in the rollouts. This is one of the reasons why other off-policy algorithms such as DDPG and SAC generally do not work well for offline RL. This is also the out-of-distribution action problem that recent offline RL algorithms like BCQ and BEAR attempt to address. Unlike MPO, during the AWR updates (Eq 10), the policy is updated using only actions observed in the replay buffer. We do not need to query a Q function with new actions that may be different from those in the replay buffer. Therefore, AWR is not susceptible to issues due to errors in the value estimation on out-of-distribution actions. This is also the main reason why AWR is able to learn effective policies in the fully offlines setting.
>
> For regular RL tasks, MPO, like other DDPG style algorithms, requires a number of stabilization techniques such as double Q-learning and target networks, none of which are needed for AWR. AWR does use some standard normalization techniques, but as we show in E.2, AWR still performs well even when these additions are disabled. AWR is therefore quite a bit simpler than algorithms like MPO.
>
> Re: SAC and TD3 on imitation tasks
> We have not been able to successfully apply SAC or TD3 on the motion imitation tasks with complex simulated agents. We are also not aware of any prior work that has demonstrated successful results with SAC and TD3 on learning these complex motion skills. While we are not certain of the reasons why those algorithms do not work on these tasks, off-policy algorithms such as SAC and TD3 tend to be more unstable than policy gradient algorithms. The AWR updates are more reminiscent of policy gradient methods than Q-learning based techniques, such as SAC and TD3, which might be one of the reasons why AWR is more effective for these tasks.

---

> > ### Comment · AnonReviewer1 · 2020-11-19
> > **Question about the policy update**
> >
> > Thank you for the response. The contribution of the paper is clarified by the response. Let me ask another question about the policy update.
> >
> > In AWR, the policies are updated using the actions stored in the replay buffer. I think that the this approach can be used to MPO and SAC because they also train a policy by maximizing the (weighted) log-likelihood. Meanwhile, the updating the policy using the actions in the replay buffer is not applicable to TD3 because it requires generating actions from the actor to backpropagate the gradient. Am I correct? Do you think that updating the policy with the action in the replay buffer in SAC and MPO improves their performance on offline RL?  If this technique improves the performance of other RL methods, I believe that the contribution is more significant.

---

> > > ### Author Response · Authors · 2020-11-20
> > > **SAC and MPO**
> > >
> > > SAC also requires backprop from the Q function to the policy, similar to DDPG and TD3, except SAC uses stochastic actions for the policy updates, which requires the reparameterization trick to compute the gradients for the policy. Therefore, using actions from the replay buffer is also not immediately applicable to SAC either.
> > >
> > > For MPO, it is indeed using a weighted maximum likelihood loss function to update the policy. So updating the policy just with actions from the replay buffer could improve performance for offline RL, since it will avoid issues due to out-of-distribution actions with a static dataset.

---

### Official Review · AnonReviewer3 · 2020-10-26
**Unclear what new knowledge is generated**

**Rating:** 3
**Confidence:** 4

**Review:**

This paper focuses on developing an RL learning algorithm that is simple and can significantly improve performance over existing algorithms. The paper presents the algorithm AWR, which is an extension of the algorithm reward weighted regression. The primary extensions of RWR are using the advantage function instead of the q function and the ability to use experience replay. Experiments on common environments are conducted to evaluate the performance of AWR and compare it to other algorithms. There are ablation experiments to justify the choice of some of the extensions.

The development of simple, effective off-policy algorithms for reinforcement learning is still an open problem, and this paper tries to take a step towards addressing it. The paper gives a detailed derivation of the proposed algorithm, and the ablation studies provide some information as to what components in the algorithm make it useful. However, I do not believe this paper is ready for publication because the extensions are minor, and experiments lack scientific rigor, making it unclear what is to be learned from the paper. There are also some errors in the paper.

There are some essential questions the paper should address but it did not.

What is the benefit of using the baseline in RWR? Does it change the weighting of the update, or is it just a variance reduction technique?

The beta term in RWR is adapted during learning. How does the choice of beta in AWR affect the weighting of rewards during learning?

The paper says there is a theoretical analysis of the algorithm, but I only see the algorithm's derivation. What analysis is this statement referring to?

The extension of RWR to using experience replay is formulated as using a mixture policy of past policies using weights w_i for each policy pi_i. How are weights w_i chosen?

The algorithm, as defined, performs a global maximization at each step. However, neural networks are used, and this maximization cannot be guaranteed. How are the updates performed? There are other changes to make the algorithm used in the experiments, but they are not connected to the algorithm's design, e.g., the normalized advantage function.

Experiments:
The primary motivation and claim of this paper is the design of a more effective algorithm. Therefore, one should expect the presented algorithm to be a significant increase in performance. However, this cannot be concluded based on the experimental methodology used. The main issues are with how hyperparameters are selected and the lack of statistical analysis of the results.

Some undefined processes set the hyperparameters for AWR, and the parameters for the other algorithms were left unspecified. Based on this experimental setup, it is impossible to tell if AWR is better or worse than the other algorithms or if it is just due to the specific setting of hyperparameters.

The performance results are given along with standard deviations over ten trials. However, this does not directly quantify how likely these results are achieved. Furthermore, it has been pointed out in several past works that the performance of RL algorithms are highly stochastic, and more trials and proper statistical analysis is needed to gain confidence in the outcomes (Colas et al., 2018, Henderson et al., 2017).

There exist more rigorous evaluation procedures that may be useful in improving this paper (Dodge et al., 2019, Jordan et al. 2020, Sivaprasad et al. 2020).

Mathematical errors:
The jump from (9) to (10) is not correct. Sampling from a fixed-sized dataset is not equivalent to sampling from the state distribution of a policy.

In the algorithm, samples of states and actions are drawn according to the data distribution. However, this does not correspond to the empirical samples of discounted state distribution; i.e., there needs to be a gamma^t term in the expectation (Thomas, 2014).


References

Colas, C., Sigaud, O., & Oudeyer, P. Y. (2018). How many random seeds? statistical power analysis in deep reinforcement learning experiments. arXiv preprint arXiv:1806.08295.

Dodge, J., Gururangan, S., Card, D., Schwartz, R., & Smith, N. A. (2019). Show your work: Improved reporting of experimental results. arXiv preprint arXiv:1909.03004.

Henderson, P., Islam, R., Bachman, P., Pineau, J., Precup, D., & Meger, D. (2017). Deep reinforcement learning that matters. arXiv preprint arXiv:1709.06560.

Jordan, S. M., Chandak, Y., Cohen, D., Zhang, M., & Thomas, P. S. (2020). Evaluating the Performance of Reinforcement Learning Algorithms. In Proceedings of the 37th International Conference on Machine Learning.

Sivaprasad, P. T., Mai, F., Vogels, T., Jaggi, M., & Fleuret, F. (2020). Optimizer benchmarking needs to account for hyperparameter tuning. In Proceedings of the 37th International Conference on Machine Learning.

Thomas, P. (2014, January). Bias in natural actor-critic algorithms. In International conference on machine learning (pp. 441-448).

---

> ### Author Response · Authors · 2020-11-16
> **Response to R3**
>
> Re: theoretical analysis:
> The theoretical analysis refers to the analysis of combining AWR with experience replay. As we show in Sec 3.2, the update procedure outlined in Alg 1 indeed optimizes the expected improvement over a sampling policy modeled by a trajectory level mixture of behavior policies as represented by the replay buffer. While experience replay has been used in a number of prior RL algorithms, few provide such an analysis of the algorithm when combined with experience. To the best of our knowledge, our analysis is the first of its kind for this style of algorithms.
>
> Re: weights w_i for each policy pi_i
> In Sec 3.2, the weights w_i corresponds to the likelihood of selecting samples from policy pi_i when sampling from the replay buffer. Therefore w_i is simply determined by the proportion of samples in the replay buffer that are recorded from pi_i. For example, if half of the total samples in the replay buffer is from pi_0, then w_0 = 0.5.
>
> Re: Mathematical errors: The jump from (9) to (10) is not correct
> Approximating the state distribution of a policy using empirical samples from a replay buffer is a very common technique used in nearly every modern off-policy RL algorithm. Examples include DQN, DDPG, SAC, MPO, only to name a few.
>
> Re: there needs to be a gamma^t term in the expectation
> Indeed, in theory the expectation should be computed with respect to the discounted state distribution, in which case the samples should be drawn from the replay buffer using geometric sampling. However, in practice the vast majority of implementations of RL algorithms simply sample uniformly from the replay buffer. We have found this simple strategy to also work well for AWR, and avoids the additional complexity required for sampling from the discounted state distribution.

---

> > ### Comment · AnonReviewer3 · 2020-11-17
> > **Corrections needed**
> >
> > Thank you for clarifying the details of the theoretical analysis. I believe the statement should be expanded in the paper to say what precisely the theoretical analysis is.
> >
> > Thanks for stating what the weights are. Can this be added to the paper?
> >
> > Regarding the jump from (9) to (10), my point is that the statement of equivalence is incorrect, not that sampling from a replay buffer is not useful.
> >
> > Regarding sampling from the discounted state distribution, I agree that it is common to sample uniformly from a replay buffer, but the algorithm as written does not match the theory. This change is acceptable but should be pointed out in the paper.
> >
> > Are there any comments to other issues I mentioned in the review?

---

### Official Review · AnonReviewer4 · 2020-10-29
**Resubmission without new work**

**Rating:** 3
**Confidence:** 3

**Review:**

This paper presents a reinforcement learning algorithm that applies advantage-weighted regression. In each iteration, it samples trajectories from a mixture of previous policies, estimates the value function and then computes the advantage value to estimate the policy. The idea is very similar to the work published in “Neumann, Gerhard and Peters, Jan R, Fitted Q-iteration by advantage weighted regression, Advances in neural information processing systems, 2009”, starting from reward-weighted regression and further developing to advantage weighted regression.  The difference in this paper is to add a constraint on the policy search, requiring the policy to be similar to the sampling policy. However, this constraint has also been studied in the paper "Christian Wirth and Johannes Furnkranz and Gerhard Neumann, Model-Free Preference-based Reinforcement Learning, AAAI 2017" (It seems not in reference). Overall, it may enhance this paper if it has more technical novelty when developing a new algorithm.

My major concern is that this paper has been submitted last year and resubmitted this year without new additions. Moreover, this paper has been available to the public since last year with information on authors and affiliations, which may violate the double-blind review policy (e.g. the version updated on Oct 7, 2019 at https://arxiv.org/abs/1910.00177).

---

> ### Author Response · Authors · 2020-11-16
> **Response to R4**
>
> Re: resubmission without new work
> We would like to point out that since the submission last year, we have added the following:
> More thorough discussion of similar algorithms in the related work section, and comparisons to prior algorithms such as LAWER and REPS (Fig 8)
> Tuned and improved performance of baseline algorithms (e.g. SAC and TD3)
> Ran experiments with more trials (10 seeds per algorithm)
> Additional ablation experiments for design decisions such as weight clipping (E.1) and normalization techniques (E.2).
> A more in-depth discussion of the similarities and differences to policy gradient algorithms (Sec D) as suggested by last year’s reviewers.
>
> Re: arxiv and double blind reviews
> As stated in the ICLR submission policy:
> https://iclr.cc/Conferences/2021/CallForPapers
> “Submission of the paper to archival repositories such as arXiv is allowed.”

---

### Official Review · AnonReviewer2 · 2020-10-29
**Interesting approach, lacks Theory.**

**Rating:** 4
**Confidence:** 4

**Review:**

This paper presents Advantage weighted regression, which relies on two sub-routines : (a) train a value function baseline using regression, and, (b) policy learning, which is advantage weighted.  The paper is well written, and has experiments on both on-policy and offline tasks, with ablation studies on various algorithm design choices.

Some questions:

[1] Contributions: it appears the objective is a minor modification off prior work, where, the returns are replaced with advantages. While this makes intuitive sense, and it does indicate improved results, the actual contribution is rather limited. While the paper does present results in off-policy settings, these derivations etc. are standard. Can the authors describe precisely what design decisions they propose in this work (on top of RWR/other prior work) that are novel?

[2] The authors mention they present a theoretical analysis (e.g. in page 1) of AWR. While I see a derivation of the AWR update, this isn’t a theory analysis of any of the proposed algorithm's behavior. In particular, can the authors make a rigorous claim as to why this is a better algorithm than current approaches? This could be through bounding the variance of the updates and indicating this improves over prior work, or, alternatively, showing rates of convergence and how it compares against that of standard NPG. Alternatively, one can view the exponentiation of the advantage as reward transformations which tend to impact the convergence, if one utilizes a policy gradient method for obtaining the new policy (see for e.g. Ghosh et al. 2020 (An operator view of policy gradient methods)). However, these need to be precisely sketched out. Without such results, the paper indicates its merit through its experiments.

[3] Experiments: Looking at table 1: I see that AWR appears to be significantly worse than SAC in several tasks - it is again unclear as to why this is considered “competitive in terms of final performance compared to prior works” as written in table 1’s captions. Furthermore, in terms of performance of TRPO, I find that the results appear to be very much lower compared to what I believe they obtain (for e.g. see Rajeswaran, Lowrey, Todorov and Kakade (Neurips 2017)) - could you mention how these experiments were conducted and why there are these discrepancies in terms of final performance?

[4] Ablation: How is the \beta parameter of AWR chosen? How sensitive is the behavior of the algorithm to this hyper-parameter?

---

> ### Author Response · Authors · 2020-11-16
> **Response to R2**
>
> 1) As shown in Figure 3, the different components of our method, such as the use of a baseline for calculating the advantages, experience replay, and TD-lambda all contribute to performance improvements over RWR. These are not techniques that have been incorporated in prior works that use RWR. We also provide an analysis of AWR when combined with experience replay (Sec 3.2). While experience replay has been used in a number of prior RL algorithms, few provide such an analysis of the algorithm when combined with experience replay. To the best of our knowledge, our analysis is the first of its kind for this type of algorithms.
>
> 2) The theoretical analysis refers to the analysis of combining AWR with experience replay. As we show in Sec 3.2, the update procedure outlined in Alg 1 indeed optimizes the expected improvement over a sampling policy modeled by a trajectory level mixture of behavior policies as represented by the replay buffer.
>
> 3) SAC does outperform AWR by a large margin on 2 of the 6 tasks we considered. But for the other tasks, AWR is within the margin of error of the SAC performance and performs better on one of the tasks.a
>
> The TRPO results are collected by running the publicly available implementation provided in OpenAI Baselines. Other papers have also reported similar figures for TRPO. Other implementations may have some differences and additions that lead to better performance.
>
> 4) As mentioned in 3.3, \beta is selected through an adaptive scheme, where \beta is set to the standard deviation of all advantage values in the replay buffer. This is analogous to the advantage normalization technique used in frameworks such as OpenAI Baselines.

---

> > ### Comment · AnonReviewer2 · 2020-11-23
> > **Response to author feedback**
> >
> > Thank you for your response.
> >
> > After going over the feedback, (i) \beta hyper-parameter ablation hasn't been shown, (ii) experimental results are reasonable but they don't offer strong conclusions in favor of AWR or why AWR is favorable despite these results, (iii) I still believe AWR has to be strengthened by working out theory using convergence analysis and characterizing variance/effects of reward transformations. Owing to these points, I will retain my original score on this submission.

---

### Decision · Program_Chairs · 2021-01-07
**Final Decision**

**Decision:**

Reject

**Comment:**

This paper aims to develop a simple yet efficient deep RL algorithm for off-policy RL. The proposed method uses advantages to as weight in regression, which is an extension of the known method of reward-weighted regression. The paper is in general nicely written, and it comes with a set of theoretical analyses and experiments. While all reviewers admit that the approach is interesting and the work makes an attempt to solve an important yet open problem, there are several aspects of the paper that make it not ready for publication in its current form:

- Novelty: As pointed out by reviewers, the proposed method appears to be a minor modification of existing off-policy solvers. Although the use of advantages as weights makes intuitive sense, it is unclear why and how the new method significantly differs from and outperforms existing methods. Going forward, it would be helpful if the authors could present more convincing arguments/experiments to demonstrate the power of ARW, relative to similar existing methods.

- Experiments provide some insights into the difference between several algorithms, but the results are not strong enough to support the claim of the paper. Please see reviewers' comments for more details. We strongly recommend the authors to take these comments into consideration and develop more rigorous experiments to demonstrate advantages of AWR.

- Theoretical analysis is limited. As R#2, R#3 mentioned, the theory analysis in the paper seems to not match the algorithm, and there remain bugs, this it doesn't add to the paper. Although theory might not be the focus of the paper, if the authors decide into include theoretical analysis, the analysis would hopefully provide insights into why and by how much the approach is better.